# Analysis of Employment Change in Response to Hurricane Landfalls

Yuepeng Cui[1], Daan Liang[2] and Bradley Ewing[3]

1. Department of Transportation, Fujian University of Technology, Fuzhou 350118, China;
2. Department of Civil, Environmental,& Construction Engineering, Texas Tech University, Lubbock 79409,USA;  3. Rawls College of Business,Texas Tech University, Lubbock 79409,USA)

*Correspondence to*: Yuepeng Cui (ypcui916@hotmail.com)

**Abstract.** Hurricanes cause extensive harm to local economies and in some cases the recovery may take years. As an adequate, skilled, and trained workforce is a prerequisite for economic development and capacity building, employment plays an important role in disaster reduction and mitigation efforts. The statistical relationship between hurricane landfalls and observed changes in employment at the county level is investigated. Hurricane impact is classified into temporary and permanent categories. In the former category, the level of economic activities is lowered following a hurricane landfall but quickly recovers to the pre-storm norm. In contrast, the permanent shift alters the mean value of the data and results in lasting losses in future years. The results show that Hurricane Katrina produced significant permanent impact on Orleans County, Louisiana. Chambers and Fort Bend counties experienced significant temporary impact due to the landfall of Hurricane Ike. The results are further discussed through qualitative analysis of various social, economic, and engineering factors in these affected communities. The findings support the notion that higher resilience level leads to quicker recovery after a disaster. However, the underlying data-generating processes are characterized and tested in a more detailed manner.

Key Words: Employment, Hurricane impact, Resilience Level, Time Series

## 1. Introduction

Natural hazards are an ongoing part of human history, which is caused by natural hazards rather than man, for example an earthquake, flood, or hurricane, and coping with them is a critical element of how resource use and human settlement have evolved (Adger 2005). It is estimated that during the period of 2006 to 2016, natural disasters affected more than 3 billion people, resulted in over 750,000 deaths, and cost more than $600 billion around the world (Hallegatte et.al 2017). Globally, 1.2 billion people, or 23% of the world's population, live within 100 km from the coasts (Nichols 2003), and the percentage is likely to increase to 50% by 2030. Many of these coastal areas have high exposure to hurricane, tsunami, earthquake, and other disasters.

Based on the statistics from Congressional Budget Office, the annualized economic losses due to hurricanes in the United States are estimated at $28 billion. The top state contributing to that sum is Florida (55

percent), followed by Texas (13 percent) and Louisiana (9 percent). Hurricane Katrina was the costliest storm by far at $160 billion (in 2019 dollars).

In the aftermath of hurricanes, disruptions to business activities, supply chains and failure of infrastructures, often results in the redistribution of resources (Chow and Elkind 2005; Kaisera, et al. 2009; Comfort and Haase 2006; Sword-Daniels, et al. 2015). The capability to produce goods and services may be lost and the natural rate of employment may drop making for higher levels of unemployment (Ewing 2009; Schulte, Tobben. et al 2015). During the subsequent recovery phase, the affected communities engage in debris cleanup and redevelopment designed to quickly restore local employment and other economic activities to pre-storm levels (Burton 2015). An increasing frequency of disasters lead to the investment deficiency and economic recession which may result in the decline of employment (NBER 2009, Klomp 2014). The process of economic recovery may require months or even years (Mel and McKenzie 2011). As an example, U.S. economic growth slowed to 2.6 percent in the quarter after Hurricane Katrina, compared to the previous quarter's 3.6 percent. Hurricane Katrina produced effect on 19% of U.S. oil production which cause the oil price to rise by $3 a barrel, and gas price reached $5 a gallon (Amadeo 2015). To date, researchers have identified several general classes of elements that could explain the connection between disaster impact and economic performance(Ewing, Kruse and Thompson 2009; Ewing, Kruse and Wang 2007; Ewing, Hein and Kruse 2006; Ewing and Kruse 2005, Tompkins 2005, Cutter, et. al. 2008).

Employment has been shown as a key driver of economic activities as well as a major social concern. Local area employment provides a measure of labor market conditions, and firms gain insight into output performance through adjusting employment to match the changes in demand (ILO 2008). In Australia, more than a week after the landfall of Tropical Cyclone Debbie 2017, and flooding are still widespread in North Queensland which caused significant effect on local economic. Due to the disruption of supply chain, local community experienced significant job and income losses(Lenzen, Malik. et al 2019). In New Zealand, worker's employment status were adversely affected by the disaster, and workers were less likely to work at the same company, most of them immigrate to the other regions of unaffected area in New Zealand (Fabling and Grimes 2016). The study focus on the 2004 Indian Ocean tsunami demonstrates the importance of employment to evaluate post-disaster recovery programme. (Jordan, Javernick-Will and Amadei 2015).

Employment is associated with the level of preparedness for disaster and ability to take proactive actions. Higher employment in a county, for example, often translates into higher resilience and quicker recovery process through purchasing insurance, and upgrading houses (Mayunga 2007; Xie et al 2014). The researches focused on analyzing the elements of vulnerability and disaster recovery highlight the importance of employment status for speeding up the recovery process after the disaster struck the community(Frazier et al 2014; Stewart et al 2014; FEMA 2018). In addition, the literature related to displacement following the landfall of hurricanes in general, suggests that employment instability is an important component of displacement. The overwhelming reason referred to by these migrants was job-

seeking or relocation due to employment (Chaganti and Waddel 2015; Sterett 2015; Meléndez and Hinojosa 2017). Therefore, examining the changes in employment following the landfall of hurricane would not only present the health of business environment but also indicate the state of broad economic recovery. Disasters also provide opportunities to study the economic dimensions of large-scale shifts.

The research presented in this paper is focused on analyzing temporary (i.e. transitory) and permanent impacts of hurricanes on affected communities. More specifically, we examine the disruption of employment and investigate the statistical relationship between hurricane landfalls and observed changes in local employment. In some counties the time series are lowered following a hurricane landfall before quickly returning to the pre-storm level. In contrast, other counties experience permanent shifts in the mean value and sustain long-lasting losses. Understanding the dynamic response of employment to hurricanes can help the local communities to assess their future risk to hurricanes and devise effective mitigation measures.

The remainder of the paper is organized as follows: In Section 2, we describe two historical hurricanes in the study. In Section 3, data specifications of employment for affected counties are presented. In Section 4, we introduce the Auto Regressive Integrated Moving Average (ARIMA) model and discuss its application to the data. Results are discussed in Section 5, and qualitative explanation of the results are described in Section 6. Concluding remarks and future extensions are given in Section 7.

## 2. Hurricanes Under Study

Hurricanes often bring highly detrimental consequences when they made landfall in urban areas (Voogd 2004). Two historical hurricanes-Hurricane Ike and Hurricane Katrina are selected in this study, because they produced big impact on densely populated areas of New Orleans, Louisiana and Houston, Texas, respectively.

On the morning of September 13, 2008, Hurricane Ike as the fourth most destructive hurricane in the United States made final landfall at Galveston island as a Category 2 hurricane with maximum sustained winds nearing 110 mph (175 km/h) and then moved onto the mainland which covered over 425 miles of Texas coastline (Berg 2009). It was the first hurricane to hit Houston area since the landfall of Hurricane Jerry in 1989. Hurricane Ike ripped through the Houston area, and the eye of the storm passing over Harris County, TX. Houston MSA as the fourth largest city in the U.S., at least 20 people died due to the landfall of Hurricane Ike. Nearly 2,900 units were deemed unfit for living, with losses exceeding $208 million. The storm led to minor damage for about 251,000 residential homes. The total damage cost was estimated around 4.6 billion (Harris County Texas 2009). According to the estimation of he U.S. Department of Energy, about 2.6 million customers experienced power failure in Texas and Louisiana. Due to the high wind of Hurricane Ike, many windows of the city's tallest building in downtown Houston had broken (Clark 2008).

Hurricane Katrina made its final landfall  as a Category 3 hurricane near the Pearl River at the
Louisiana/Mississippi border. Hurricane Katrina's high wind combined with its enormous size at landfall
caused the tremendous storm surges along the Gulf Coast area. The hurricane severely impacted or
destroyed business buildings and residential homes in New Orleans and some other heavily populated
areas(NOAA 2005 and USGS 2008). Approximately 80% of New Orleans flooded, and the depth of the
flood is up to 20ft following the landfall of hurricane. The total economic damage from Hurricane Katrina
is around $160 billion (in 2019 dollars), nearly two times the cost of the previously most expensive
hurricane, Hurricane Andrew (USDC 2006).
**3. Data Specification for Hurricanes and Employment**
A brief introduction of the data used in the empirical analysis and some initial observations for the entire
hurricane periods will be introduced in this section. The Hurricane-relevant parameters such as wind speed,
central pressure and radius were considered as important atmospheric factors for assessing and predicting
the physical damages caused by hurricanes (Zhang and Wang 2003). Storm parameters data are obtained
from the National Hurricane Center (NHC) for two hurricanes including latitude, longitude, wind speed and
pressure. Sample storm track data about Hurricane Katrina and Hurricane Ike are shown in Table 1. In
addition to physical damage, hurricanes also pose a risk to local employment market and economic
situation(Zhang et al., 2008).

122                    Table 1 Historical hurricane tracks for Hurricanes Ike (2008) and Katrina (2005)

The population in New Orleans declined from over 400,000 to near zero in less than a week after Hurricane
Katrina swept the Gulf of Mexico (Vigdor 2008). The number of layoff events in Louisiana and Mississippi
increased greatly and rapidly in September 2005 soon after Hurricane Katrina (USBS 2006). The number
of workers and the number of firms operating in New Orleans were also reduced. The subsequent rebuild
process was hindered by absent employees as many of them had homes destroyed or their family required
urgent care. It's previously reported that employees who experience injury from the disaster may be more
likely to be absent from work in the weeks following the event (Byron and Peterson 2002). In September
2005, Mickey Driver, a spokesperson for Chevron stated, "we are trying to find out where they've (our
employees) gone, what their current situation is and what we can do to help them". The organization's
ability to recover from the disaster can be weakened due to the lack of employee access to work (Kroll. et
al 1991). Monthly employment data for the counties within Houston MSA and New Orleans MSA are
obtained from the Bureau of Labor Statistics (http://www.bls.gov).

135           Figure 1 Monthly employment time series in Orleans County before and after Hurricane Katrina

Figure 1 shows the monthly employment time series in Orleans County. The red 'X' marker denotes the
month in which Hurricane Katrina made landfall. The MSA lost more than 80,000 jobs (or 33%)
immediately after Katrina, gained some back during the initial one month of recovery, and then lost again
during the recession. Casual observation indicates that Hurricane Katrina was a contributing factor
responsible for such a major reduction in employment.

141        Figure 2 Monthly employment time series in St.Charles County before and after Hurricane Katrina

Figure 2 presents the historical monthly employment data in St. Charles County. It is clear at first glance
that the storm led to an initial drop in employment (2000 jobs or 8%) but the magnitude wasn't as severe as
Orleans County. The ensuing trajectory was also markedly different, enjoying a long expansion after the
Great Recession.
Figure 3 Monthly employments time series in five counties within Houston MSA before and after
Hurricane Ike
Figure 3 presents the historical monthly employment data for five counties within the Houston MSA. Again,
the red 'X' marker denotes the month when Hurricane Ike made landfall. Comparing to Hurricane Katrina,
it is not apparent whether or not Ike led to a drop in employment as the five counties appear to have been in
the midst of a decline (or period of slowing growth) preceding the storm. However, it does appear that
there is an abatement in cyclical behavior (i.e. volatility) in the post-storm period and perhaps even an
uptick in Brazoria County.
**4. Methodology for Quantifying Hurricane Impact**
The ARIMA (Auto-Regressive Integrated Moving Average) model of time series mainly include three
parameters $p$, $d$, and $q$. The process of determining the integral numbers of auto-regressive $p$, integrated $d$,
and moving average $q$ could identify the patterns of the model. It generally started with finding accurate
value of parameter $d$ because it provides important information about the order of time series being
investigated. $P$ is the number of auto-regressive terms that describes the number of lag observations
included in the model. For example, in a model with three auto-regressive terms ($p=3$) indicates that the
current date observation depends on three previous period observations. The value of $q$ represents the
moving average term which is only related to the random errors that occurred in past time periods. For
example, a model with one moving average term suggests that the current date observation is determined
by the preceding random shock to the series. If a parameter equals to a value of 0, which indicates to not
use that element of the model.
Two common unit-root tests are implemented to test the stationary of the respective time series and to
identify the value of $d$ in the model. Phillips and Perron (1988) and the Augment Dickey-Fuller (ADF)
tests are applied in our study to analyze the stationary of employment variables in different counties. $d$
equals to 0 indicates that time series is stationary in levels, if not, the first(or second, third....) difference of
the time series will be examined until the time series is shown as stationary time series data.
The results of the ADF unit root test suggest that each series of employment in different counties is non-
stationary in levels, but it is stationary in the first difference. PP unit root test presents the same result of the
ADF test. Therefore, the first difference of each sequence is used as input to identify ARIMA model in
order to compare the results of each county. Box-Jenkins methodology (Maddala 1992) is involved in the
identification and estimation of ARIMA (*p*,1,*q*) which applies partial auto-correlations and auto-
correlations of stationary time series data to obtain the best fit of time series data. The values of p and q is
determined by choosing the minimum value of Akaike information criterion(AIC).
ARIMA model with intervention analysis is mainly applied to estimate the impact caused by specific
external event such as natural hazards, policy change ,etc (Enders 2009). Baade and Baumann (2007) use
ARIMA model with intervention analysis to estimate the Hurricane Andrew impact on taxable sales in the
respective cities. This technique has been widely used in many fields of research studies ranging from
evaluating the impact of the financial crisis on Nigeria crude oil export(Adubis and Jolayem 2015) to assess
the effects of Federal Emergency Management Agency (FEMA) policies change on employment in
hurricane-stricken cities (Ewing and Kruse 2005). Intervention analysis offer a formal test to evaluate
several patterns of distortions (changing the mean function or trend) as a result of external shock.
Table 2 presents the result of ARIMA model selection based on standard Box-Jenkins methodology with
Akaike information criterion. Consequently, the first difference in each series are used as input to identify
the values of *p* and *q* in ARIMA model, thus the results of hurricanes impact on different counties can be
compared.
190                                     Table 2 ARIMA model selection

Intervention analysis is carried out in the following steps. We first identify the ARIMA model for each
county before the month of hurricane landfall. A binary (intervention) variable with a value of 1 or 0 is
defined as a intervention variable, where a value of 1 flags the hurricane periods (either the month of
hurricane at landfall or entire post-hurricane period accordingly) and takes the value zero at other times.
Then, the model with intervention variable is re-estimated for the whole time series data (i.e., pre- and post-
hurricane period). The effect of hurricanes on employment can be understood by examining the magnitude
and statistical significance of coefficients on intervention variables.
Two types of intervention variables are added to the ARIMA model separately to evaluate the hurricane
impact on the employment at the county level. The "temporary" impact of hurricane may be captured by
the intervention variable that equals one in the month of hurricane landfall and zero at other times. The
"permanent" effect of the hurricane may be modeled by the intervention variable that equals one since the
month of hurricane landfall through the end of the sample period and zero elsewhere. Note that the latter
represents changing mean or trend in the growth rate of employment. Equation (1) shows the ARIMA
model with intervention analysis.

$$\Delta y_t = c + a_1 \Delta y_{t-1} + ... + a_p \Delta y_{t-p} + e_t + b_1 e_{t-1} + ... + b_q e_{t-q} + \beta D \qquad (1)$$
where $D$ is the intervention variable (i.e., temporary or permanent), $\beta$ is the associated coefficient, and c is
a constant term, $p$ is the number of lags on the auto-regressive term, $a_1,...a_p$ are the coefficients for AR
model, and c is constant, $b_1,...,b_q$ are the coefficients of the MA part in the model.
There are several points worth paying attention on ARIMA intervention model. The design of the ARIMA
intervention method focuses on the time series relationship between a specific variable and an event
(especially the time period of the occurrence of the events) and isolates the effects of changes in time series
behavior of the variable before and after the event. In addition, an appropriate defined ARIMA model can
achieve this without adding additional control variables, and these variables are effectively handled in the
error term (Enders 2009). Excessive specification (i.e. adding irrelevant or statistical redundant control
variables) leads to multi-collinearity, and standard errors often result in  lower accuracy in the time series
models. Therefore,  diagnostic tests are conduct on residual errors to determine that 1) they perform well
(normal, constant variance) and 2) the error items do not contain additional information that can be used to
improve the prediction accuracy of the model. In generally, ARIMA model has the ideal characteristics
with less and  better error terms. Results for the temporary effect are presented in Table 3, and the
permanent effect results are shown in Table 4. Statistical significance at the 5% level is indicated by "**".
The adjusted R-square represents the extent of the total variance of the dependent variable which can be
explained by the independent variable, and estimated number of independent variables  are also considered.
The adjusted R-squares reported in Table 3 are fully within the acceptance range of the model specified in
the first difference. The F-statistic tests the null hypothesis that all coefficients except the constant term are
equal to zero.The results of F- statistics shown in the tables below indicate that the null hypothesis is
rejected, which prove the rationality of the existence of the model.
Hurricane Ike produced significant temporary impact in Chamber County and Fort Bend County as the
employment growth rate slowed down by 8.2% in Chambers County, and 4.3% in Fort Bend County. In
contrast, permanent change in the mean growth rate is found to be significant in Orleans County where the
mean growth rate slows down by 8.6%.

232                 Table 3 Results of temporary impact for employment


234                 Table 4 Results of permanent impact for employment

Figures 4 and 5 further illustrate the temporary and permanent impacts that hurricanes have on
communities. The shaded area in these figures represents the post-storm period. Actual and forecast values
are shown as well as the (one standard deviation from the mean) upper and lower bounds for the forecast
(or confidence bands). The temporary reduction from Hurricane Ike occurred in Chambers County where
employment dropped by 8.1% but recovered within two years (see *Figure 4*) when the series re-entered the
areas shown within the confidence bands. In contrast, it took Orleans County about 7 years (2005 through
2012) to return to the pre-storm employment level following Hurricane Katrina. These two cases present a
clear difference in time scale in how local employment recovered from hurricanes.
Furthermore, others have founds that long term recovery from disasters usually takes three-five years
(Webb, Tierney and Dhlhamer 2002, Fussell 2015 and Marks 2015). Therefore, we define the threshold for
permanent effect in this study as 3 years or longer. In other words, if it takes 3 years or more for
employment to return to within the forecast confidence bands, the impact will be considered permanent.
Otherwise, it will be considered as temporary impact.
Figure 4 Temporary effects of Hurricane Ike in Chambers County
Figure 5 Permanent effects of Hurricane Katrina in Orleans Parish County
Further investigations are conduct in relation to the changing tendency of various types of employment in
Houston MSA and New Orleans MSA following the landfall of hurricanes. Monthly employment extracted
from Bureau of Labor Statistics in construction, retail sale, whole sale and utilities industries of Houston
MSA and New Orleans MSA are shown in Figure 6 and Figure 7 below. The shaded ares represent the post
hurricane period, the 'X' indicates the month that Hurricane Ike or Hurricane Katrina made landfall.
Figure 6 Monthly Employment in four industries of Houston MSA

The construction employment in Houston MSA increased slightly immediately following the landfall of
Hurricane Ike. And the employment in the other three industries (retail sale, whole sale and utilities)
present the decreasing tendency following the landfall of Hurricane Ike, and employment in retail sale,
whole sale and utilities show increasing tendency until the beginning of 2010(which is one year and half
after the landfall of Hurricane Ike).
Figure 7 Monthly Employment in four industries of New Orleans MSA
Unlike the employment in Houston MSA, the employment in four industries of New Orleans MSA present
a huge drop following the landfall of Hurricane Katrina immediately. Only employment in utilities show a
long-term increasing tendency starting from 2007 which is 2 years after the landfall of Hurricane Katrina.
Employment in whole sale, retail sale and construction has a short term quick increase, then it presents a
fluctuation trend, among them employment in whole sale and retail sale are not even back to the pre-
disaster level until 2019.
5. Qualitative Explanation of the results
Based on the analysis above, Hurricane Ike produced significant but temporary impact on employment in
Chambers County while Galveston, Harris and Brazoria counties didn't experience any significant impact.
Then the question is raised: what has contributed to a community's ability to withstand and recover from
disaster?
We attempt to address this question through the prism of resilience. Disaster resilience is defined as the
capacity or ability of a community to anticipate, prepare for, respond to, and recover quickly from impacts
of disaster (Foster 2006). According to Walker et al. (2006), adaptability is mainly controlled by all forms
of capital, the number of government and institutions in the system. The capitals of the system are
fundamental components for the resilience study of the entire community, e.g., social, human, economic,
physical and natural, which are referred to as elements of resilience. The evaluation of community
resilience is a complex process due to the dynamic interactions among people, community, society, and
environment (Foster 2006; Pelling et. al. 2019). Several indicators have been applied to assess the
community resilience under each element of resilience are shown in Table 5.

Table 5 Framework of evaluating resilience (Mayunga 2007)

Hurricane Ike made a direct hit in Galveston but failed to produce any significant impact on its employment.
A possible explanation for this is that while Galveston County is highly susceptible to hurricanes and
tropical storm-force winds, it has experienced several hurricanes in the past and may have adapted
accordingly (e.g. Hurricane Alicia in 1983, Hurricane Allison and Hurricane Jerry in 1989). Several
emergency studies suggest that community resilience could be built through the adoption of social
media(Dufty 2012). Through warning system, the community could promote effective action to respond to
disaster (Tasic and Amir 2016). Thanks to advanced weather monitoring systems, the National Hurricane
Center (NHC) predicted correctly that Hurricane Ike would hit the Galveston (FEMA 2008). This triggered
a mandatory evacuation for Brazoria (located to the south of Galveston County) and Galveston Counties.
Residents who followed the order took necessary steps to protect themselves, their families and properties.
As a result, residents in these two counties by and large were better prepared for Hurricane Ike than those
living in other counties. Morss and Hayden (2010) interviewed 49 residents affected by the landfall of
Hurricane Ike, and performed approximately five weeks after the landfall. Ninety percent of interviewees
said they prepared their residences before the landfall of Hurricane Ike. Only five reported that they don't
prepare specifically for Ike. However, all five residents who did not prepare suffered heavy loss, mostly of
which were caused by flooding. This further supports the discussion that better preparation could enhance
the resilience of affected counties.
Harris County, the biggest county within Houston MSA, has a highly diversified economy. Cutting edge
technologies allow the energy industry to continue to power the Houston region's growth, while research
and development breakthroughs regularly occur at the world's largest medical complex - The Texas
Medical Center – which adds to regional prosperity. Besides, it has a growing population represented by all
major racial/ethnic groups. Harris County's well-developed financial infrastructure, skilled workforce,
good labor relations and diverse population attracts many international companies. All of these factors in
turn could be responsible for raising its capability to resist external shocks like Hurricane Ike and recover
more quickly in the aftermath.
Unlike Harris County, Chambers County is very rural and a population of just over 26,000. Hurricane Ike
damaged its utilities and critical infrastructures, including power lines, substations, and water and sewer
plants. The estimated loss was $12.1 billion (TEES 2009). At the same time, the storm disrupted many of
its economic engines, including the University of Texas Medical Branch (UTMB), the ports and waterways,
agricultural and natural resources, and the tourist industries (USHUD 2009). The University of Texas
Medical Branch (UTMB) at Galveston recorded an employment decline during this time, largely due to the
effects of Hurricane Ike, which damaged several buildings.
According to Abel et al. (2006), the ability to self-organize is the foundation of resilience. A need exists for
local systems to be interconnected and connected to a larger, national system in order to deal with
disturbances. It is also important that these local networks maintain self-reliance, or the ability to subsist
without the larger system (Baker and Refsgaard 2007). This can be accomplished through establishing trust
among the population through networks and institutions, their leaders, and the information disseminated to
the community (Nkhata et al. 2008, Longstaff and Yang 2008). Building network is an essential element in
disaster reduction, and resilience level of a community heavily depend on the established network of people
from different sectors(Chatterjee, Ismail and Shaw 2016). Collaboration among networks can greatly
improve resilience of a community.The management method frequently taken by the New Orleans
government was a command and control approach that targeted a specific variable and reduced resilience
by ignoring other parts of the system (Gunderson 2009).
Lastly, it's worth noting that the hurricane's impact doesn't permeate all elements of a community on an
equal basis. Previous analysis of the same two hurricanes on building permits (Cui, Liang and Ewing 2015)
reveals that significant temporary impact was evident in Orleans, Chambers, Fort Bend, Harris, Liberty and
Montgomery while significant permanent impact was evident only in St.Charles. We suggest that three
counties - Orleans, Chambers, Fort Bend – were least resilient among their peers and suffered the most
during these two hurricanes.
**6. Concluding Remarks and Future Research**
The results from this empirical study illustrate the impact of hurricanes on local employment. An
interesting finding is that, regardless of storm, the effects are limited to either being temporary or
permanent in nature. In the temporary impact case, the level of employment is lowered following a
hurricane landfall but quickly recovers to the pre-storm norm. In contrast, the permanent impact shifts the
mean value of the time series data and persists for a longer period of time. The results may be explained
through five forms of capital used to evaluate the resilience of an affected community. The comparison
among communities identifies strengths and weakness in these various forms of capital and their
contribution to recovery. Understanding the empirical results in the context of social, economic, human,
physical and natural capital provides local officials with insight and possible actions to ensure the outcomes
can be significantly improved.
Hurricane Harvey highlights the idea that people are a critical link in the effort to build community
resilience (Savio 2018). Business owners need to form a recovery plan in which several aspects of human
capital are considered. For example, could employees continue working safely during recovery? Can they
work remotely? Are they trained in disaster preparedness? For businesses relying on local customers, will
they be able to access goods and services?
Future work in this area of study should target two main unresolved issues. The first one is to examine
employment across different demographic groups stratified by income, age, race, etc. at the local scale,
which is critical for planning, mitigation and recovery from hurricanes. The goal is to identify the
distributional and disproportionate impacts of hurricanes in various sub-populations so that policies and
programs could be tailored for their specific needs. The second issue is to improve our understanding of
fundamental factors and underlying processes of disaster recovery. To that end, we need to extend the
analysis to other socioeconomic settings. For example, a cross-country panel data set can be used to
analyze critical drivers of community resilience in developed and developing countries.
The methodology presented in this paper could be considered as an entry point to addressing the complex
problems related to disaster resilience. Focused, limited-scope empirical studies like ours play a major role
in bridging the knowledge gaps and catalyzing innovations.

**Author Contribution**
Yuepeng Cui are responsible for model development, calculation, plot figures and writing. Daan Liang are
responsible for data specification part and revising the manuscript. Bradley Ewing are responsible for
giving guidance about model development, result discussion and conclusion.
**Data availability**
The data are publicly accessible, the description of the data are present in the Data Specification section.
**Competing interests**
The authors declare that they have no conflict of interest.
**Acknowledgement**
This material is partially based upon work in part supported by the National Science Foundation
under Grants CMMI-1000251 and CMMI-1131392. Any opinions, findings, and conclusions or
recommendations expressed in this paper are those of the authors and do not necessarily reflect the
views of the National Science Foundation.

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

**Table 1: Historical hurricane tracks for Hurricanes Ike (2008) and Katrina (2005)**

| Date/Time | Longitude | Latitude | Wind Speed(kt) | Pressure(mb) |
|---|---|---|---|---|
| Hurricane Katrina | | | | |
| 26/1800 | 24.9 | 82.6 | 85 | 968 |
| 27/1200 | 24.4 | 84.7 | 100 | 942 |
| 28/1200 | 25.7 | 87.7 | 145 | 909 |
| 29/0600 | 28.2 | 89.6 | 125 | 913 |
| Hurricane Ike | | | | |
| 10/1800 | 24.2 | 85.8 | 85 | 958 |
| 12/1800 | 27.5 | 93.2 | 95 | 954 |
| 13/1200 | 30.3 | 95.2 | 85 | 959 |
| 14/1200 | 37.6 | 91 | 40 | 987 |



**Table 2: ARIMA model selection**

| Hurricane Name | County | ARIMA Model | Adjusted R-Square | F-statistic |
|---|---|---|---|---|
| Hurricane Katrina | Orleans | (0,1,3) | 0.672650 | 28.05442 |
| | St. Charles | (1,1,3) | 0.548294 | 18.19573 |
| Hurricane Ike | Brazoria | (2,1,3) | 0.302821 | 11.41402 |
| | Chambers | (2,1,3) | 0.362174 | 12.12940 |
| | Fort Bend | (0,1,2) | 0.534298 | 12.91547 |
| | Galveston | (2,1,2) | 0.428823 | 15.30493 |
| | Harris | (1,1,2) | 0.478316 | 28.94065 |



**Table 3:Results of temporary impact for employment**

| Hurricane | County | Temporary | | Adjusted R-square | F-statistic |
|---|---|---|---|---|---|
| | | P-value | Beta | | |
| Hurricane Katrina | Orleans | 0.8609 | 0.005476 | 0.521029 | 47.76391 |
| | St. Charles | 0.7781 | -0.003473 | 0.274538 | 7.856402 |
| Hurricane Ike | Brazoria | 0.3020 | -0.001221 | 0.416745 | 31.35297 |
| | Chambers | 0.0000** | -0.081789** | 0.342465 | 15.52769 |
| | Fort Bend | 0.0387** | -0.043339** | 0.350011 | 19.28911 |
| | Galveston | 0.65491 | -0.217338 | 0.318773 | 18.22978 |
| | Harris | 0.18665 | 0.001188 | 0.256785 | 9.798675 |


**Table 4:Results of permanent impact for employment**

| Hurricane | County | Permanent | | Adjusted R-square | F-statistic |
|---|---|---|---|---|---|
| | | P-value | Beta | | |
| Hurricane Katrina | Orleans | 0.0000** | -0.08653** | 0.5692541 | 30.89562 |
| | St. Charles | 0.2882 | -0.003649 | 0.387652 | 10.76492 |
| Hurricane Ike | Brazoria | 0.3020 | -0.001221 | 0.386158 | 19.22739 |
| | Chambers | 0.3942 | -0.003558 | 0.257711 | 10.99645 |
| | Fort Bend | 0.1407 | -0.002233 | 0.278219 | 15.99100 |
| | Galveston | 0.9467 | -0.003265 | 0.378517 | 19.06807 |
| | Harris | 0.2271 | -0.057741 | 0.339228 | 20.68832 |





**Table 5: Framework of evaluating resilience (Mayunga 2007)**

| Element of resilience | Indicator of resilience | Explain |
|---|---|---|
| Social Capital | Trust, Norms and Networks | Facilities coordination and cooperation Facilities access to resources. |
| Economic Capital | Income, savings and investment | Reduces poverty Increases capacity e.g. insurance speeds recovery process |
| Human Capital | Education, Health Skills Knowledge/Information | Increase knowledge and skill to understand community risks Increase ability to develop and implement risk reduction strategy |
| Physical Capital | Housing, Public facilities, business/industry | Communication and transportation evacuation |
| Natural Capital | Resources stocks, land and water ecosystem | Sustains all forms of life Increase protection to storms and floods Protects the environment |



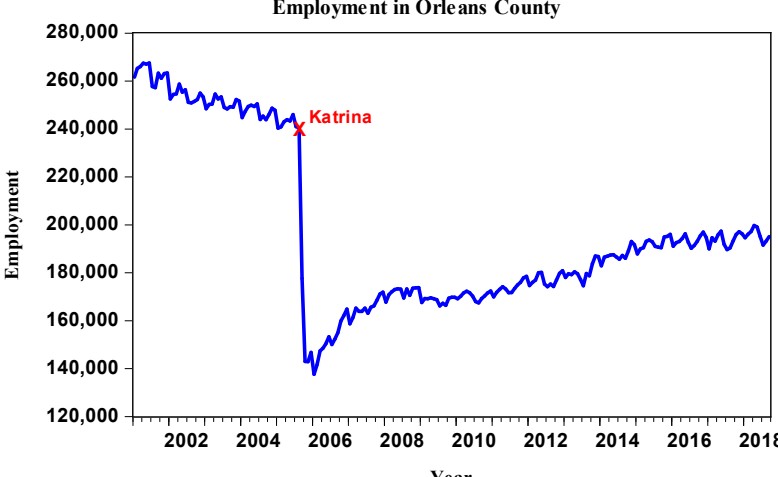


**Figure 1: Monthly employment time series in Orleans County before and after Hurricane Katrina**

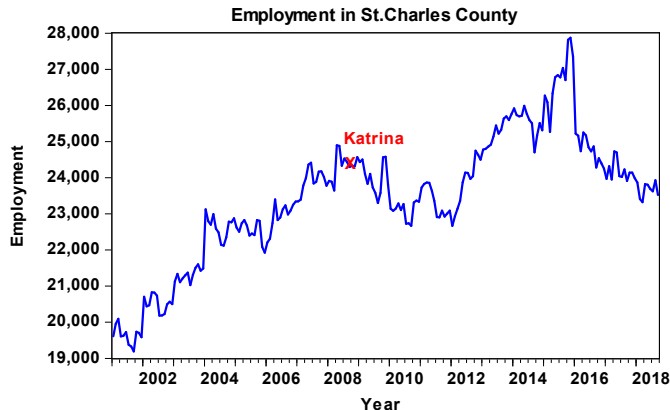


**Figure 2: Monthly employment time series in St.Charles County before and after Hurricane Katrina**

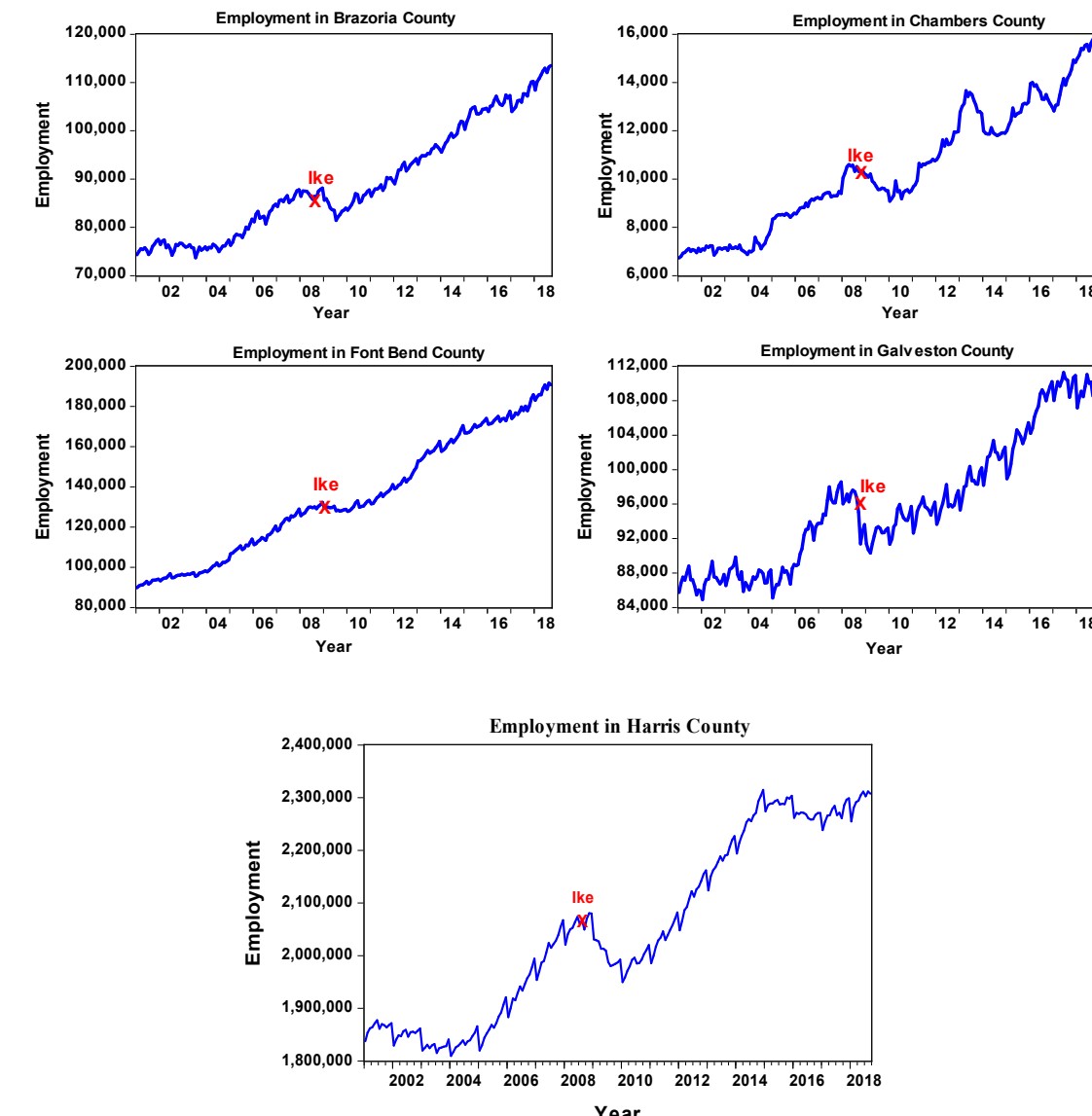



**Figure 3: Monthly employment time series in five counties within Houston MSA before and after Hurricane Ike**

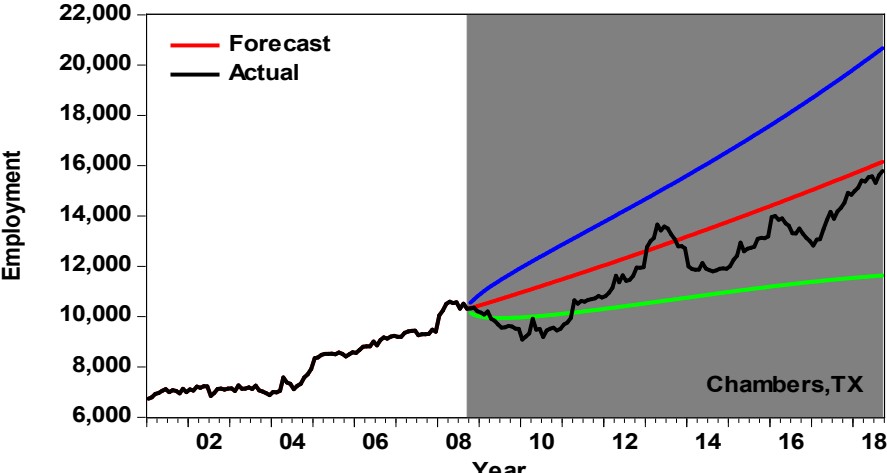


**Figure 4: Temporary effects of Hurricane Ike in Chambers County**

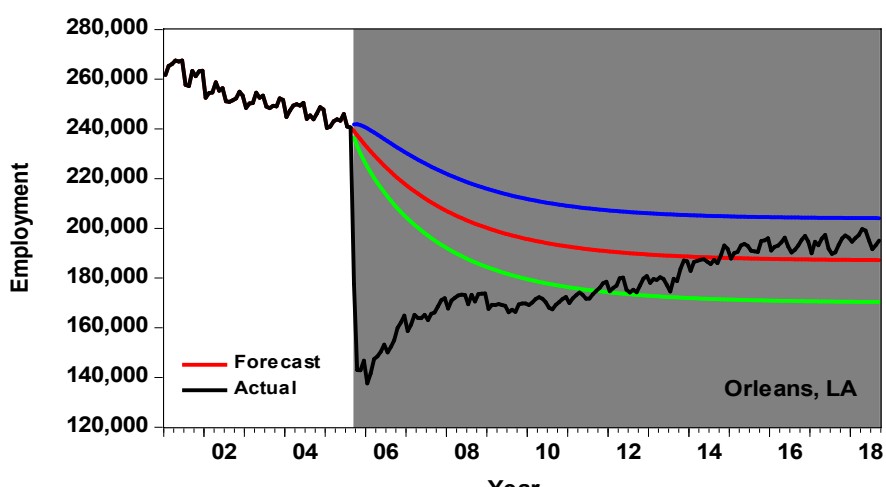


**Figure 5: Permanent effects of Hurricane Katrina in Orleans Parish County**

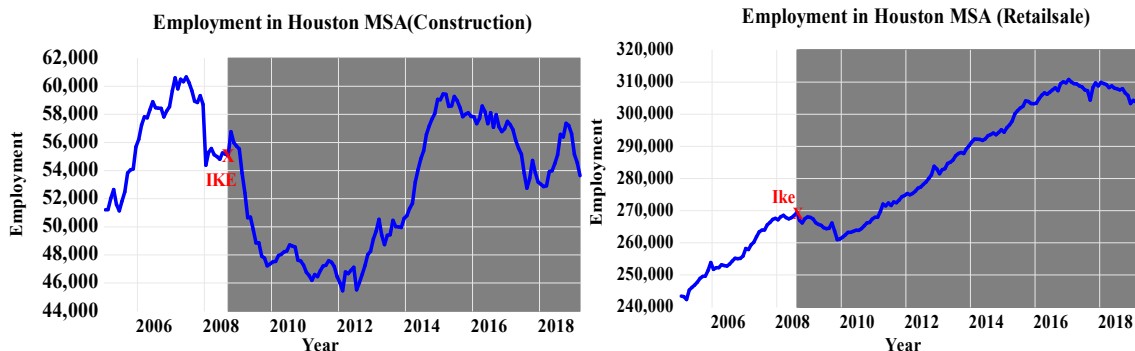


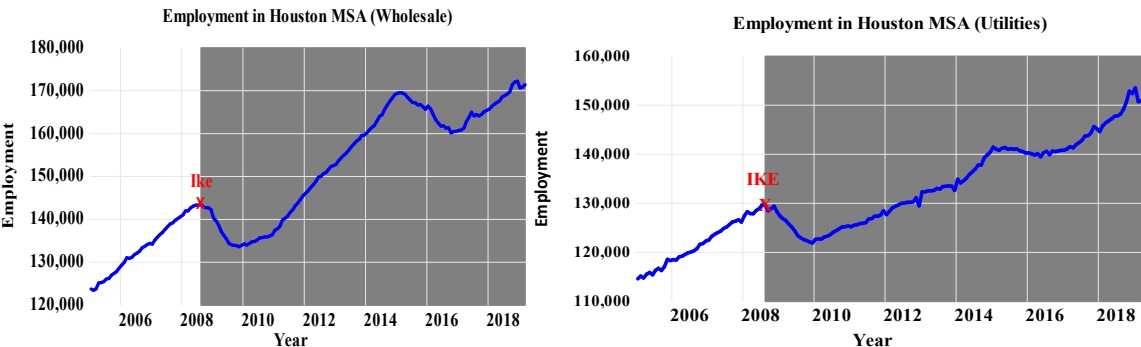


531                     Figure 6 Monthly Employment in four industries of Houston MSA




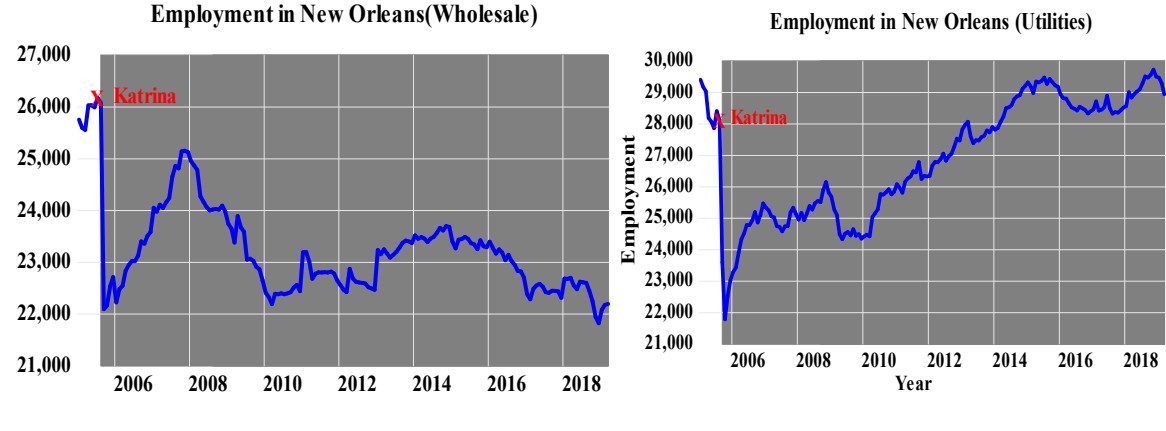


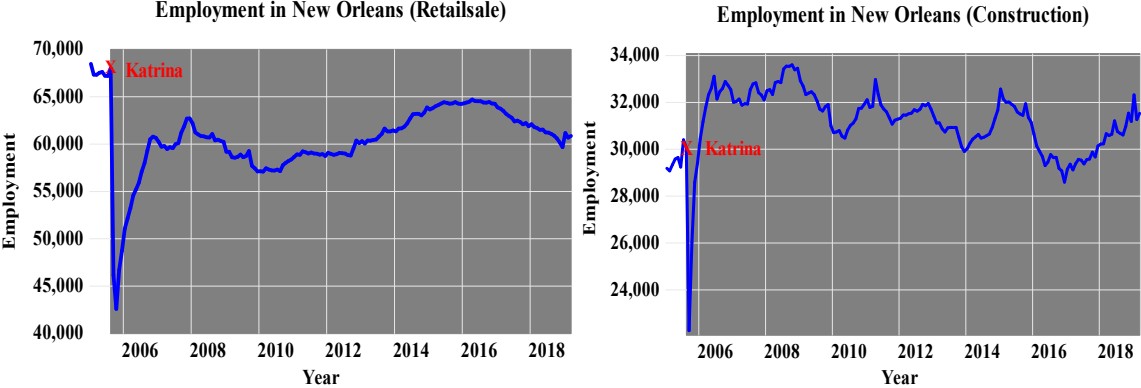

536                     Figure 7 Monthly Employment in four industries of New Orleans MSA
