# Peer review of "Analysis of Employment Change in Response to Hurricane Landfalls"

_Natural Hazards and Earth System Sciences, 2019_

## Referee Comment (RC1) · Anonymous Referee #1 · 18 Jul 2019

This paper discussed an interesting question about how 'large' natural hazards events influence national/federal economy. In overall, I have some questions/suggestions about your paper: ->in overall: you missed large amount of current published papers in this field, especially some of your quoted papers are some years old, please, can you update and add broader literature review in your paper in this field? Especially also work conducted outside of the US. ->on page 2 line 41/43: you make the comparison between the impact of Hurricane Katrina on national economic growth: can you add a reference and further elaborate why you observe this slow down, show the economy of Louisana and Mississippi that important role in the overall economy of the United States? Or was it more the large oil spills caused by Hurricane Katrina in the Gulf of Texas. ->on page 4 line 119-136: why you observe this temporary change:

[Figure]

because of bankruptcy of businesses, lack of insurance payments or because of labor market system in terms of unemployment or because of the businesses were temporary closed? Can you elaborate this in a more detail and provide also some details about the US labor market also in terms of unemployment benefits? Can you also provide a more in-depth overview, which sectors were mainly cause this temporary shortfall (service sector, productive sector etc.), was this more likely for large businesses or small-medium enterprises, can you also provide a more geographical overview where the unemployment rate increased after the event. Can you also observe any changes in the local consumption, house price, inflation, credit, debt rate? ->on page 7 line 211: a major question is: which type of jobs got lost (based on which economic sector and how this sector is regulated/organised in the US?). ->page 9: discussion is somehow missing, please link your results with other international references? ->in overall can you observe any changes in the long-run growth rate in your model/examples?

Minor point: Don't use the term natural disasters term "natural disaster" is a misnomer. Disasters such as hurricanes and earthquakes result from a combination of natural hazards and social and human vulnerability. Calling them 'natural disasters' artificially naturalises the harms they cause.

---

## Short Comment (SC1) · 19 Aug 2019

**Referee Report**

Analysis of Employment Change in Response to Hurricane Landfalls

Natural Hazard and Earth System Sciences

NHESS-2019-201

**Overview**

This paper conducts an analysis of how employment responded to large hurricane events and finds that in some cases, Hurricane Katrina in Orleans Parish, the effects can be long lasting but in others, like the Houston area after Hurricane Ike, effects are short term.  This analysis is a good addition to the literature and should be published.  However, this paper needs some work before acceptance.  My comments are listed below in two categories; 1) writing and 2) needed work on the qualitative section.

**Writing**

For most of the piece the writing is fine but there are enough instances of awkward language, missing punctuation and grammar that it is sometimes difficult to read.  Below is a list the line numbers of sentences that need attention.

Line numbers:  40, 60, 69, 79, 89, 94, 152, 164, 200, 202, 207, 226, 250, 253

**Qualitative Section (Lines 233 – 292)**

First, shouldn't this section be separate from Section 4?  Next, this is the weakest piece of the author's argument.  I don't see it as a qualitative analysis of resilience in the aftermath of both Katrina and Ike but simply a recitation of information provided by other entities.  Studies which interviewed residents after the storms and compared their assessment would make a better case than listing how the difference in warnings made residents better prepared.  Then the statement is made that Fort Bend County is deficient in natural and human capital.  Based on what?  Beginning on line 278 the authors quote Abel et al (2006) about the ability to self-organize is an important element in resilience.  Great point.  How can they show that this happened in some places and not others without citing or conducting a qualitative study to show that?

---

## Author Comment (AC1) · 2 Oct 2019

| Reviewer #1 | Response |
| --- | --- |
| 1. you missed large amount of current published papers in this field, especially some of your quoted papers are some years old, please, can you update and add broader literature review in your paper in this field, Especially also work conducted outside of the US. | We have quoted a number of additional papers published between 2011 and 2019 some of which addressed disasters outside the U.S. For example, Tropical Cyclone Debbie (2017) caused widespread flooding in North Queensland, Australia and the affected communities experienced significant job and income losses. The analysis of 2017 Hurricane Maria's impact on Puerto Rico focused on the displacement and job-seeking. |
| 2. on page 2 line 41/43: you make the comparison between the impact of Hurricane Katrina on national economic growth: can you add a reference and further elaborate why you observe this slow down, show the economy of Louisiana and Mississippi that important role in the overall economy of the United States? Or was it more the large oil spills caused by Hurricane Katrina in the Gulf of Texas. | A few references have been added to further explain the importance of the oil production to the national economy. It stated in one paper that "Hurricane Katrina produced effect on 19% of U.S. oil production which cause the oil price to rise by \$3 a barrel, and gas price reached \$5 a gallon." |
| 3. on page 4 line 119-136: why you observe this temporary change:because of bankruptcy of businesses, lack of insurance payments or because of labor market system in terms of unemployment or because of the businesses were temporary closed? Can you elaborate this in a more detail and provide also some details about the US labor market also in terms of unemployment benefits? Can you also provide a more in-depth overview, which sectors were mainly cause this temporary short-fall (service sector, productive sector etc.), was this more likely for large businesses or small-medium enterprises, can you also provide a more geographical overview where the unemployment rate increased after the event. Can you also observe any changes in the local consumption, house price, inflation, credit, debt rate? | This section serves as an initial observation of employment changes in selected counties in the aftermath of hurricanes. The distinction between temporary and permanent impacts will be determined by the ARIMA model analysis in the subsequent section.

We explained the results through the prism of community resilience based on five capitals -social, economic, human, physical and natural. Some of your suggestions can be incorporated. |
| 4. on page 7 line 211: a major question is: | We will add more figures based on business |

| | |
|---|---|
| which type of jobs got lost (based on which economic sector and how this sector is regulated/organized in the US?). | dynamics statistics to illustrate the employment changes by sectors and firm sizes. |
| 5. page 9: discussion is somehow missing, please link your results with other international references? | The explained part is called "Qualitative Explanation of the results". Now we list it as Section 5. We also add some international references to explain the results. For example, "Building Disaster Risk Reduction in Asia- A Way Forward ADPC Looks Ahead to 2015", Combining Disaster Risk Reduction, Natural Resource Management and Climate Change Adaptation in a New Approach to the Reduction of Vulnerability(2013, Canada) |
| 6. In overall can you observe any changes in the long-run growth rate in your model/examples? | In this study, Hurricane Katrina was shown to produce permanent impact on Orleans Parish County. References were added to define the threshold for permanent effect as 3 years or longer. |
| 7. Don't use the term natural disasters term "natural disaster" is a misnomer. Disasters such as hurricanes and earthquakes result from a combination of natural hazards and social and human vulnerability. Calling them 'natural disasters' artificially naturalises the harms they cause. | We use the term of "Natural Disasters" throughout this paper to differentiate from man-made disasters (e.g. terrorist attack). A sentence was added at the beginning to clarify. |

---

## Author Comment (AC2) · 2 Oct 2019

| Short Comment | |
|---|---|
| 1. Qualitative Section (Lines 233 – 292) First, shouldn't this section be separate from Section 4? | As suggested, Qualitative Section will be renumbered to 5. |
| 2. Qualitative Explanation of the results, this is the weakest piece of the author's argument. I don't see it as a qualitative analysis of resilience in the aftermath of both Katrina and Ike but simply a recitation of information provided by other entities | We have cited past researches on the factors contributing to community resilience and supporting the result of our ARIMA models. |
| 3. Studies which interviewed residents after the storms and compared their assessment would make a better case than listing how the difference in warnings made residents better prepared. | We will add this point to our paper. |
| 4. Then the statement is made that Fort Bend County is deficient in natural and human capital. Based on what? | The contracting evidences between the ARIMA model and reported major power outage led to the speculation of deficiencies in natural and human capitals. This has been removed. |
| 5. Beginning on line 278 the authors quote Abel et al (2006) about the ability to self-organize is an important element in resilience. Great point. How can they show that this happened in some places and not others without citing or conducting a qualitative study to show that? | As the part of future work, we could conduct a survey on how organizations of a community adapt to hurricanes. We could also investigate whether the community has specific programs to promote hazard communication. Same can be done to businesses affected by hurricanes. |
| Line numbers: 40, 60, 69, 79, 89, 94, 152, 164, 200, 202, 207, 226, 250, 253 | We have corrected these errors. Your suggestions are much appreciated. |

---

## Author Response (AR1)

| Referee Comments | Response |
|---|---|
| 1. you missed large amount of current published papers in this field, especially some of your quoted papers are some years old, please, can you update and add broader literature review in your paper in this field, Especially also work conducted outside of the US. | We cited a number of additional papers published between 2011 and 2019 some of which addressed disasters outside the U.S. For example, Tropical Cyclone Debbie (2017) caused widespread flooding in North Queensland, Australia and the affected communities experienced significant job and income losses. The analysis of 2017 Hurricane Maria's impact on Puerto Rico focused on the displacement and job-seeking. Refer to Lines 54-62, and 65-75. |
| 2. on page 2 line 41/43: you make the comparison between the impact of Hurricane Katrina on national economic growth: can you add a reference and further elaborate why you observe this slow down, show the economy of Louisiana and Mississippi that important role in the overall economy of the United States? Or was it more the large oil spills caused by Hurricane Katrina in the Gulf of Texas. | A few references were added to further explain the importance of oil production to the national economy. It stated in one paper that "Hurricane Katrina produced effect on 19% of U.S. oil production which cause the oil price to rise by $3 a barrel, and gas price reached $5 a gallon." Refer to Line 45- 48. |
| 3. on page 4 line 119-136: why you observe this temporary change:because of bankruptcy of businesses, lack of insurance payments or because of labor market system in terms of unemployment or because of the businesses were temporary closed? Can you elaborate this in a more detail and provide also some details about the US labor market also in terms of unemployment benefits? Can you also provide a more in-depth overview, which sectors were mainly cause this temporary short- | This section serves as an initial observation of employment changes in selected counties in the aftermath of hurricanes. The distinction between temporary and permanent impacts was later determined by the ARIMA model analysis in the subsequent section. We explained the results through the prism of community resilience based on five capitals -social, economic, human, physical and natural. Your suggestions were fully considered. |

| | |
|---|---|
| fall (service sector, productive sector etc.), was this more likely for large businesses or

small-medium enterprises, can you also provide a more geographical overview where

the unemployment rate increased after the event. Can you also observe any changes

in the local consumption, house price, inflation, credit, debt rate? | |
| 4. on page 7 line 211: a major question is: which type of jobs got lost (based on which economic sector and how this sector is regulated/organized in the US?). | We plotted additional figures based on employment data in Houston and New Orleans MSA to illustrate the employment changes in various industry sectors. Refer to Line 251-275 |
| 5. page 9: discussion is somehow

missing, please link your results with other international references? | "Qualitative Explanation of the Results" was relabeled as Section 5. We also added some international references to explain the results. Refer to Line 294- 297, and 328-330. |
| 6. In overall can you observe any changes in the long-run growth rate in your model/examples? | In this study, Hurricane Katrina was shown to produce permanent impact on Orleans Parish County. References were added to define the threshold for permanent effect as 3 years or longer. |
| 7. Don't use the term natural disasters term "natural disaster" is a misnomer. Disasters such as hurricanes and earthquakes result from a combination of natural hazards and social and human vulnerability. Calling them 'natural disasters' artificially naturalises the harms they cause. | We used the term of "Natural Disasters" throughout this paper to differentiate from man-made disasters (e.g. terrorist attack). A sentence was added at the beginning to clarify. Refer to Line 24. |

| Short Comment | Response |
|---|---|
| 1. Qualitative Section (Lines 233 – 292)

First, shouldn't this section be separate from Section 4? | As suggested, Qualitative Section was renumbered to Section 5. |
| 2. Qualitative Explanation of the results, this is the weakest piece of the author's argument. I don't see it as a qualitative analysis of resilience in the aftermath of both Katrina and Ike but simply a recitation of information provided by other entities | We cited prior research on the factors contributing to community resilience and supporting the result of our ARIMA models. |
| 3. Studies which interviewed residents after the storms and compared their assessment would make a better case than listing how the difference in warnings made residents better prepared. | We addressed this point. Refer to Line 295-300. |
| 4. Then the statement is made that Fort Bend County is deficient in natural and human capital. Based on what? | The contracting evidences between the ARIMA model and reported major power outage led to the speculation of deficiencies in natural and human capitals. This point was removed due to the lack of clear evidence. |
| 5. Beginning on line 278 the authors quote Abel et al (2006) about the ability to self-organize is an important element in resilience. Great point. How can they show that this happened in some places and not others without citing or conducting a qualitative study to show that? | As the part of future work, we could conduct a survey on how organizations adapt to hurricanes. We could also investigate whether the community has specific programs to promote hazard communication, business continuity. |
| Line numbers: 40, 60, 69, 79, 89, 94, 152, 164, 200, 202, 207, 226, 250, 253 | Your suggestions are much appreciated. We corrected these errors. |